# Heterogeneous wireless networks with mobile devices of multiple interfaces for simultaneous connections using Fuzzy System

**Jorge Amaro de Sarges Cardoso**[1]*, **Fabio Pereira Ferreira da Silva**[1], **Tássio Costa de Carvalho**[1], **José Jailton Henrique Ferreira Junior**[1], **Nandamudi Lankalapalli Vijaykumar**[2,3], **Carlos Renato Lisboa Francês**[1]

**1** Electrical Engineering Graduate Program (PPGEE), UFPA–Federal University of Pará, Guamá, Belém, Pará, Brazil, **2** Federal University of São Paulo (UNIFESP), São José dos Campos, São Paulo, Brazil, **3** National Institute for Space Research (INPE), São José dos Campos, São Paulo, Brazil

* jorgeamaro@ufpa.br

## Abstract

With the exponential increase in heterogeneous wireless networks today, there has been a growing interest from the academic community for issues related to handover problems. The main objective of this paper is to evaluate the quality of service and performance of a device with a dual interface that connects simultaneously to two heterogeneous networks with no competition in the exchange of packets between them. It is a proposal to solve the mitigation of handover impacts. The tool used for evaluation was the Network Simulator 2. The results showed a better use of the band in comparison to the scenario using a traditional mobile device.

## Introduction

The current demand for mobile large band undergoes unprecedented growth, especially with the emergence of a substantial number of smart portable devices. This trend puts enormous pressure over cellular communication systems for capacity requirements, Quality of Service (QoS), Quality of Experience (QoE) and energy efficiency owing to the development of applications with high bandwidth requirements, such as streaming videos and sharing multimedia files [1].

According to a survey carried out by the Global System for Mobile Communications Latin America (GSMA), almost half of the global population (3.8 billion people) is currently a mobile internet user, with a forecast of reaching 61% (5 billion) by 2025. Among the technologies, 4G was the dominant mobile technology used worldwide during 2019, accounting for more than half (52%) of global connections. Despite the increase in 5G technology, 4G will continue to expand in the coming years, reaching 56% of connections by 2025. According to the projections by GSMA, smartphones are expected to represent four out of five mobile internet connections by 2025 [2].

According to [3], the estimated worldwide increase of mobile data traffic for 2021 will be 49 exabytes. The Visual Networking Index (VNI) report by CISCO predicts that the increase of

**Data Availability Statement:** All relevant data are within the paper and its Supporting Information files.

**Funding:** The authors received no funding sources (financial or material support) for the study.

**Competing interests:** NO authors have competing interests The authors have declared that no competing interests exist.

global IP traffic for mobile and fixed networks will keep on expanding until 2022 and a significant part of this traffic will correspond to multimedia content, representing 82% of this data.

The current infrastructure has shown to perform well for voice traffic; however, as the profile of service utilization by users is changing and the use of data traffic is drastically increasing, this infrastructure will be unable to satisfactorily cope up with the new demand.

With the increase in converging services among networks of different access technologies and the evolution of telecommunications networks, the development of mobility management mechanisms and handover algorithms is of crucial importance. Such mechanisms are generally implemented in the network core or at its edge, in order to reduce the impacts caused by the migration of mobile devices between heterogeneous wireless networks. These devices are manufactured with multiple interfaces of wireless network technologies.

Mobile devices, especially smartphones, are commercialized already equipped with more than one interface for connection to different wireless networks. But technology of such devices, by default, prioritize the connection of the IEEE 802.11 standard technology over other wireless network technologies, such as 4G. Therefore, the user's mobile device is unable to use two wireless network technologies simultaneously, which could be useful for taking full advantage of the broadband available through these networks and for example, distributing the user applications between different wireless network technologies and avoiding competition from those applications used by the same wireless network interface.

Another important point is that the connection of the mobile device in a given wireless network is traditionally made by the higher signal strength, which is not always the best option, due to the network quality parameters such as: network congestion, packet collisions, packet loss, high energy consumption, and regardless of whether the network offers the minimum QoS and QoE requirements.

The main objective of this paper is to propose a mobile device that contains multiple interfaces of wireless network technologies with the capacity of multiple connections and simultaneous use of heterogeneous wireless network interfaces, in order to obtain the maximum bandwidth and a better wireless network throughput, so that there is no competition between user applications on the same wireless technology interface as the mobile device. Consequently, there is a need to make a decision on selecting a connection and this decision making will make use of the Fuzzy System.

The paper is organized as follows: section II discusses the published literature involving the use of multiple wireless technology interfaces; section III describes the environment of heterogeneous wireless networks with simultaneous connections; section IV describes the results obtained with the simulations performed and V presents our conclusion.

## Materials and methods

### Related work

This section discusses other research publications related to the new heterogeneous architectures that support the requirements of transparent vertical mobility and also how their use can effectively provide benefits.

[4], endorse the view that the proliferation of intelligent mobile services performed by the media has led to a rapid expansion in mobile data traffic, which has increasingly overloaded cellular networks and degraded the quality of services for mobile users. Therefore, effectively meeting these growing demands in a timely and economical way becomes a critical challenge for network operators.

The authors report that, recently, a new paradigm called Dual Connectivity has been gaining strength through 3GPP standardization activities and industrial practices. Dual connectivity

allows mobile users to communicate simultaneously with a macrocell Base Station (BS) and with an Access Point (AP) via two distinct radio interfaces.

In the same paper the authors propose a new unloading traffic mechanism from mobile equipment. Such mechanism takes into account co-canal type interferences on mobile devices when transferring data to the AP and aims to minimize the total cost of the system. Results demonstrate that the algorithm proposed by the authors can achieve the optimal solution with a significantly reduced computational time. However, the research does not propose a mechanism to avoid competition between the interfaces of the device itself.

Another study [5], employs Dual Connectivity as a technical solution for the problem of increasing handovers in heterogeneous networks with the purpose of minimizing the average delay of the system, subject to a restriction in the probability of blocking background and foreground users. Although the paper presents a handover management mechanism, it does not show a concurrency algorithm between the device own interfaces, which is capable of blocking calls to reduce network saturation.

[6], consider a heterogeneous network architecture, in which the devices are equipped with two wireless interfaces, allowing the simultaneous transmission and reception of packets. The first interface communicates with the base station using long-range wireless technology (for example, LTE) and the second interface communicates with nearby devices using short-range wireless technology (for example, IEEE 802.11 ad hoc mode).

The architecture proposed by the authors uses a handover delay reduction heuristic based on graph theory, which uses instantly decodable network coding. The paper does not show any vertical handover management mechanism; it takes into account exclusively static, non-mobile scenarios that do not carry out QoE metrics evaluation.

In another paper [7], the researchers propose a resource allocation and inter-cell interference management scheme for small cells with integrated licensed and unlicensed band interfaces to coordinate and combine dual interfaces of cellular mobile technologies (for example, LTE or 3G) and non-cellular technologies (for example, Wi-Fi or White Space TV) simultaneously in the same device.

The authors formulate an optimization problem that jointly allocates resources over both licensed and unlicensed bands with the goal of maximizing sum small cell user equipment (SUE) rate while achieving fairness among these user equipment and controlling inter-cell interference to neighboring macrocell users. The formulation of the proposed optimization problem as an efficient and low complexity linear programming.

Although Mobile devices do connect to multiple wireless network interfaces simultaneously, there is no proposed algorithm for transparent vertical handover management, as well as there is no evaluation using QoE metrics.

In [8], the authors address the dissemination of content based on network encoding in a dual interface mobile network. The article considers a cellular network with a base station (eNodeB) and N mobile devices. The devices are close to each other, which have the same demand for common content (for example, a video). The devices use two interfaces: the cellular link to connect to the base station and the Wi-Fi link to exchange the data received through device-to-device (D2D) communication.

The main contribution of the article is to provide an analytical solution to roughly calculate the average time to complete the content delivery process. In the analysis, it was considered the correlation between the coded packets transmitted, as well as the average time that mobile devices dispute to successfully transmit a packet, however the article does not do a QoE analysis to assess whether the multimedia content was transmitted with good quality for users.

In [9], the authors propose a mobility management algorithm to perform handovers efficiently between 4G and 5G radio access technologies. The proposed algorithm uses the dual

connectivity (DC) technique, as DC has less handover interruption compared to conventional handover. The mobility management scheme suggests an appropriate data sharing mechanism between 4G and 5G networks. The algorithm efficiently handles the radio access technologies.

In the authors' proposal, a two-tier network was considered, an existing 4G layer providing general coverage and a newly implanted 5G layer. This structure consists of an algorithm that considers a variety of parameters, such as cell density, coverage area, number of users, and radio channel status, to make handover decisions within the same layer (intra) or in different layers (inter).

In the DC configuration, users maintain connections to 4G and 5G networks to reduce interruptions in data transfers during handovers. The proposed structure is based on a probabilistic model that suggests an application-specific data sharing mechanism for a mobile user. The system was modeled as a Markov decision process (MDP).

Although the authors' paper presents a mobility management mechanism to perform handovers, it does not show a simultaneity algorithm between the device's own interfaces, which can block calls to reduce network saturation, does not divide user applications between the technologies of heterogeneous networks and does not analyze the quality of experience concerning users' multimedia applications.

In [10], the authors propose a Dual Connectivity scheme to millimeter wave communication (mmWave) in which user equipment (UE) is connected with two mmWave Base Stations (BSs) and a BS LTE. The authors compared the technique applied by them in the paper based on a double connection with the scheme based on a single connection through the ns-3 simulation. The simulation results show that the technique used by the authors in the paper surpasses the scheme based on a single connection in terms of QoS, network overhead, spectral efficiency, and TCP performance.

The authors' proposal helps to decrease the handover rate in an ultra-dense cellular network, which contributes to a significant reduction in the network control overhead, however, the authors do not use any artificial intelligence technique to aid in the simultaneous connection of the User Equipment (UE) or the UE handover process for new BSs and do not divide the user's application between heterogeneous wireless network technologies. The work evaluates the performance of QoS in its simulations and does not study QoE in its simulation scenarios.

Therefore, none of the articles provides a joint approach that involves a solution for the simultaneous use of heterogeneous wireless network interfaces with the distribution of user applications between these interfaces and with QoE support. QoE. Tables 1 and 2 compares the works related to the current proposal.

## Heterogeneous wireless networks with simultaneous connections

Regardless of technology, heterogeneous wireless networks offer several connectivity opportunities for mobile users, leaving them the choice on the best alternative for each situation among the available options. It is recommended that the decision on best connection choice should be made mainly based on parameters of Quality of Service and/or Quality of Experience.

Under certain circumstances, due to the mobility of the mobile user, it is necessary to change to another access point. This process is called handover. When switching access points involving different technologies, the procedure is known as vertical handover. (Fig 1).

As demonstrated in Fig 1, the mobile device is within the coverage area of two access points (Wi-Fi and 4G) requiring, necessarily selecting (normally with the best signal strength) one of the two technologies for connection. As a consequence, the mobile device is unable to explore

**Table 1. Related works (Part 1).**

| Proposal | QoE | Simultaneous Connection of Heterogeneous Networks | Avoids Concurrency of Applications on the Same Wireless Network Interface | Proposal | Proposal Focus |
|---|---|---|---|---|---|
| [4] | No | Yes | No | Dual Connectivity and Algorithm for offloading traffic from mobile equipment. | Co-channel interference on mobile devices when transferring data to the AP and aims to minimize the total cost of the system. |
| [5] | No | No | No | Dual Connectivity and Algorithm for Handover Management. | Dual Connectivity as a technique to solve the problem of increasing handovers in heterogeneous networks in order to minimize the average delay of the system. |
| [6] | No | Yes | No | Graph theory-based coding Instantly decodable network encoding. | Heuristic of handover delay reduction based on graph theory using instantly decodable network coding. |
| [7] | No | Yes | No | The formulation of the proposed optimization problem as an efficient and low complexity linear programming. | Formulate an optimization problem that jointly allocates resources over both licensed and unlicensed bands with the goal of maximizing sum small cell user equipment (SUE) rate while achieving fairness among these user equipments and controlling inter-cell interference to neighboring macrocell users. |
| [8] | No | Yes | No | Device to device communication (D2D), content dissemination based on network encoding on a dual interface mobile network. | Analytical solution to roughly calculate the average time to complete the content delivery process. In the analysis, the correlation between the coded packets transmitted was considered, as well as the average time that mobile devices. Dispute to successfully transmit a packet. |
| Current Proposal | Yes | Yes | Yes | Fuzzy system for handover management. | Allow the mobile device to be connected simultaneously to two heterogeneous wireless networks. |

the maximum available resources that the technologies offer; especially if it is using more than one application (received through a single interface). Based on all the foregoing considerations, it appears to be that there is a need to propose a mobile device with multiple interfaces that could connect simultaneously to two (or more) heterogeneous wireless networks. It should provide users with maximum use of the network throughput and thus, avoid possible competition of the applications in a single interface, as it commonly occurs with the current mobile devices.

In Fig 2 one can observe a scenario, in which the mobile device is also within the coverage area of two distinct technologies. In this new context, however, the mobile device will simultaneously connect to the two heterogeneous access points. Therefore, there is a distribution of

**Table 2. Related works (Part 2).**

| [9] | No | No | No | Dual connectivity and algorithm for handover management between 4G and 5G radio access technologies | It consists of an algorithm that considers a variety of parameters, such as cell density, coverage area, number of users, and radio channel status, to make handover decisions within the same layer (intra) or in different layers (inter). The system was modeled as a Markov decision process (MDP). |
|---|---|---|---|---|---|
| [10] | No | Yes | No | Dual Connectivity to millimeter wave communication (mmWave) in which user equipment (UE) is connected with two mmWave Base Stations (BSs) and a BS LTE | Decrease the handover rate in an ultra-dense cellular network, which contributes to a significant reduction in network control overhead, comparison of the technique based on a double connection with the scheme based on single connection through the ns-3 simulation and QoS evaluation. |

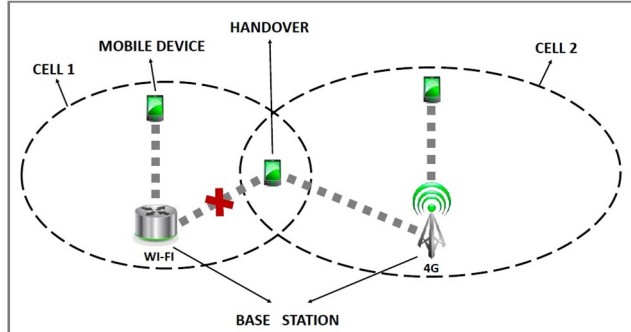

**Fig 1. Transparent vertical handover.**

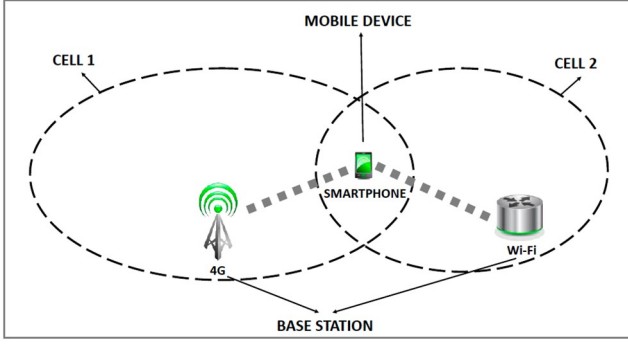

**Fig 2. Dual interface device.**

the applications (one for each interface) avoiding competition between them, and consequently, increasing the total throughput of the mobile device.

The mobile device will be connected simultaneously to two heterogeneous wireless networks. However, the mobile device will continue to identify new available access points for a possible handover to be performed. According to the proposal, the handover will not disconnect the mobile device from one technology to connect to another, since the second interface will also be in use in another connection. In this new scenario, the mobile device will only perform handover for the first interface, without impairing the active connection to the second interface.

The mobile device will be connected simultaneously to 4G and Wi-Fi networks and to prevent competition between technologies. The device performs a handover from 4G to 4G (without interfering with the Wi-Fi connection) or performs a Wi-Fi handover to Wi-Fi (without interfering with the 4G connection). Therefore, three situations are expected to occur in this new scenario: 1) Stay connected to 4G and perform handover only to Wi-Fi; 2) Stay connected to Wi-Fi and perform handover only to 4G; 3) Carry out simultaneous double handover from 4G to 4G and from Wi-Fi to Wi-Fi. In each of the three situations, the mobile device will always be connected simultaneously to both technologies to guarantee the maximum possible throughput to the user.

## Fuzzy System

Unlike traditional logic, which works with exact values, fuzzy logic allows for a level of uncertainty. The use of Fuzzy Systems has advantages such as being easily understandable, as it is

flexible, can model nonlinear functions, is based on natural language, and therefore can be built based only on the experience of specialists. Fuzzy systems are systems based on knowledge or rules. Their main feature is the knowledge base, the so-called IF-ELSE rules, which are characterized by pertinence functions. The starting point for building a fuzzy system is to obtain a collection of IF-ELSE rules from knowledge-based human experts.

A fuzzy set has an associated membership function that defines for each element of the universal set a degree of membership that can vary within the real range [0.1]. The value obtained from the membership function for a given element shows how much that element belongs to the set. In this way, an element may belong more to a fuzzy set than another.

In this paper, the Fuzzy System considers three input metrics: 1) Mobility level; 2) Intensity of the Received Signal; 3) Percentage of packets received.

The mobility level tends to indicate how long a mobile device remains within the coverage area of one cell. The faster the device, the less time it will remain in the cell, and the slower the device, the longer it will remain in the cell. The RSSI (Received Signal Strength Indication) indicates when the mobile device is likely to disconnect from an access point or not and the percentage of packages is an indicator of the quality of the service/experience offered.

The Mobility Level was divided into three sets: low (0–6m/s), average (4–11m/s) and high (above 10m/s). The Received Intensity index is a factor used to assess how likely a mobile device is to disconnect from one access point (if the signal strength is weak) and to connect to another access point (if the received signal strength is high). For this paper, it adopted and divided the received signal strength into three levels: low (-120/-100dBi), average (-115/-66 dBi), high (above -72dBi).

The Percentage of Packets Received index will determine the degree of service quality of the network, which means, the more packets discarded when connecting to a network, the worse the performance this network will offer to the mobile node.

In this paper, the percentage of packages received was classified into three sets: low (0–40%), average (30–70%), and high (above 60%), and the three indexes mentioned in the previous paragraphs, represent the input basis of the fuzzy system.

The output of the fuzzy system indicates to the mobile device whether or not it should perform the handover process, thus, the output of the fuzzy system is divided into four categories: NO, PROBABLY NO, PROBABLY YES, and YES. The inference value determines the output of the Fuzzy Systems, the handover procedure will happen when the inference value is equal to or greater than 0.6 (Fig 3).

The decision indicated by the Fuzzy System will be made according to the set of rules defined by means of empirical analysis (a series of repeated simulations). The set of rules of the

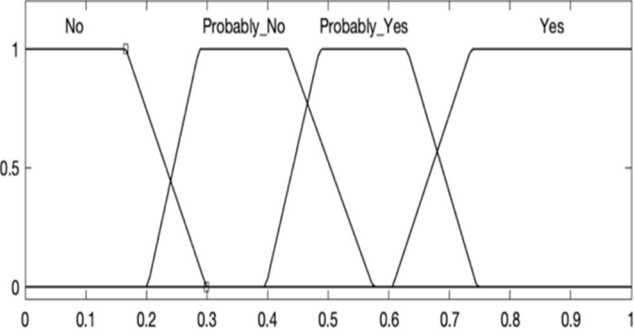

**Fig 3. Fuzzy System sets.**

**Table 3. Fuzzy System rules.**

| Mobility | RSSI | Packages | Output |
|---|---|---|---|
| Low | Low | Low | Yes |
| Low | Low | Medium | Yes |
| Low | Low | High | Yes |
| Low | Medium | Low | Probably Yes |
| Low | Medium | Medium | Probably Yes |
| Medium | Low | Low | Yes |
| Medium | Low | Medium | Yes |
| Medium | Low | High | Probably Yes |
| Medium | Medium | Low | Probably Yes |
| High | Low | Medium | Probably Yes |
| High | Low | High | Probably Yes |

fuzzy system to perform the handover is presented in Table 3 (other situations not shown in Tables 1 and 2 indicate a strong tendency of the mobile device not to perform the handover). In general, users with low mobility, or low signal strength, or a low number of packages are more likely to perform handover.

The fuzzy system will keep both connections simultaneously active, but it will also monitor the networks detected by the mobile device. At each new connection detected, the fuzzy system will evaluate and indicate whether or not the mobile device should perform the handover procedure. The fuzzy system aims to avoid unnecessary handover process and even the ping pong handover (disconnection followed by reconnection to the same network). High mobility users only perform handover in critical cases (since they tend to remain for a short time within the cell's coverage area).

Fig 4 shows the tendency to start, or not, the handover process. The blue region of the graph corresponds to a mobile device with high mobility, high RSSI, and a high number of received packets. Under these conditions, the mobile device will not perform the handover to the new network. The yellow region of the graph corresponds to a mobile user with low mobility, low RSSI, and a low number of received packets. Under these different conditions, the mobile device will perform the handover to the new network.

## Major contributions

One of the main contributions of the paper is the simultaneous use of multiple interfaces and connections on a mobile device (feature not available on current mobile devices). The proposal

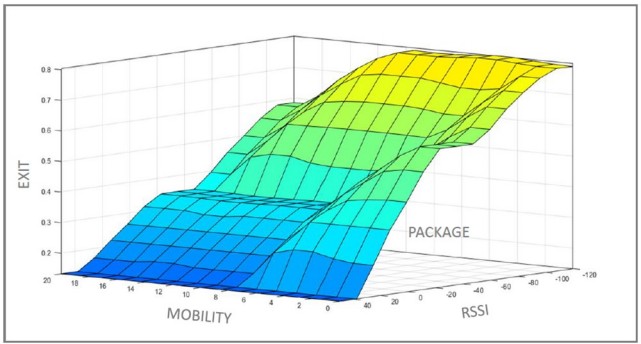

**Fig 4. Tendency to start, or not, the handover process.**

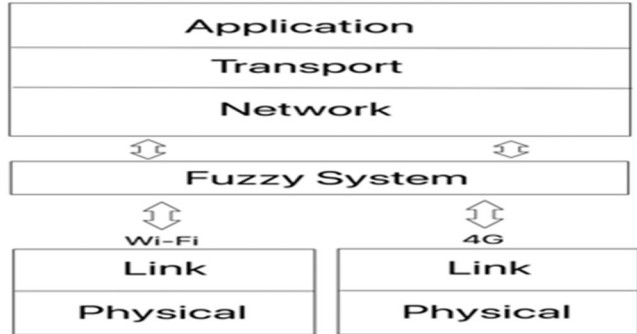

**Fig 5. Fuzzy System layer.**

of the article will act as an intermediary between the two interfaces of the mobile device, the Fuzzy System will be an intermediate layer of the physical layer collecting information from the two interfaces (Fig 5).

It is important to highlight that the Fuzzy System will balance the network by distributing applications between the interfaces and, as far as possible, equally distributing the applications between the interfaces (1 application for Wi-Fi/1 application for 4G, 2 applications for Wi-Fi/2 applications for 4G, . . .). Obviously, when there is an odd number of applications, load balancing will not be evenly distributed between the interfaces.

In this context of Next Generation Networks, the goal is to provide 100% connectivity availability, so the mobile device will have connectivity options from different technologies. Fig 6 shows the scenario in which the mobile device will be connected simultaneously to the 4G network and the Wi-Fi network and, as it moves, it will perform a handover, maintaining simultaneous connectivity (Fig 6).

The proposal, in addition to simultaneous connectivity, aims at the automatic selection of the network following the entry criteria of the Fuzzy System, thus the proposal prevents unnecessary handovers, as well as avoiding ping-pong handover (connecting to a new network and then return to the previous network). The proposal allows seamless handover, that is, the user will disconnect from the current access point only after establishing a connection with the new access point so that there is no damage to the quality of service and experience.

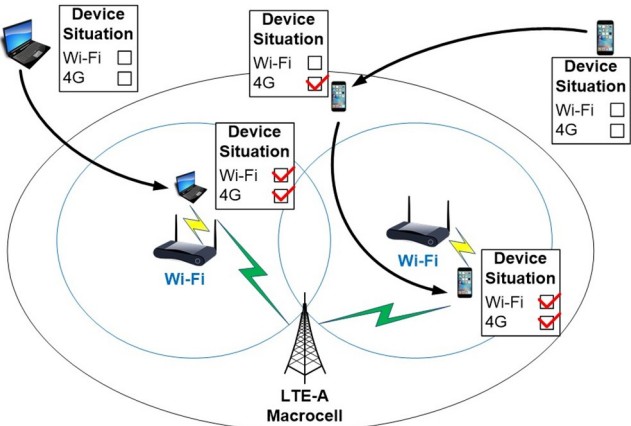

**Fig 6. Scenario multiple connection.**

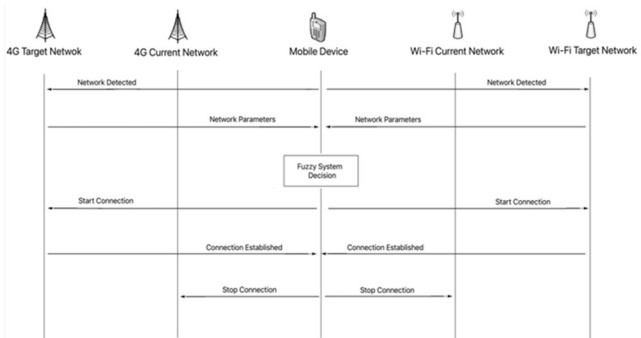

**Fig 7. Signaling handover.**

When the mobile device detects the presence of a new network, it will receive the connection parameters of the new access point, after the message the Fuzzy System on the mobile device will evaluate whether it will perform the handover or not, if the decision is yes, the device mobile will start the connection with the new access point, after the connection established with the new network it will disconnect from the current network (Fig 7).

# Results

## Performance evaluation

This section describes the methodology and metrics used for evaluating two scenarios with two types of mobile devices: the traditional one, which represents traditional mobile devices such as cell phones, and a new mobile device with a dual interface, whose purpose is to be connected simultaneously to two heterogeneous networks, with no competition between networks in their respective interfaces (as shown in Fig 8).

The performance evaluation was carried out through simulations using the Network Simulator 2. In the simulation scenarios, mobile devices had random mobilities and displacements. All the results represent an average of 100 simulations.

For the performance evaluation of the scenarios, the throughput rate was used as a quality of service (QoS) metric and, for the user's quality of experience (QoE), the following metrics were analyzed: Peak Signal to Noise Ratio—PSNR, Structural Similarity Index Method—SSIM and Video Quality Metric—VQM. Table 4 shows the simulation parameters used.

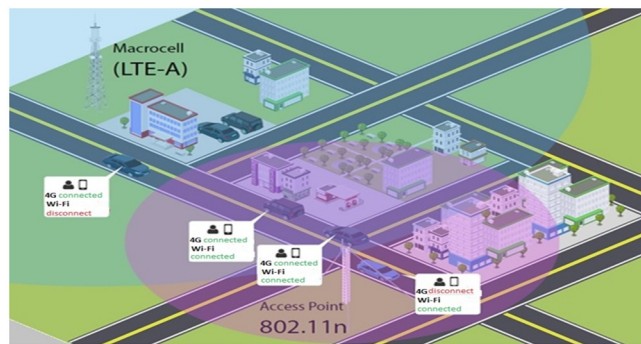

**Fig 8. Scenarios for device with dual interface.**

**Table 4. General simulation parameters.**

| PARAMETER | VALUE |
|---|---|
| Simulation Area | $2Km^2$ |
| Number of Stations | 2 |
| Number of Mobile Devices | 2 |
| Channel Wireless | Wi-Fi/LTE |
| Rate | 54 Mbps/124 Mbps |
| Power Transmission | 50 m/500 m |
| Frequencies | 2.4 Ghz/3.5 Ghz |
| Applications | CBR/Video |
| Package Size | 1024 KB |
| Simulation Time | 120 s |
| Quantity of Simulation | 100 |

## CBR application scenarios

In the scenarios using the two CBR (Constant bit Rate) applications, the traditional and the proposed devices (simultaneous use of the two wireless network interfaces), started the simulation in a region where the 4G coverage area intersects with Wi-Fi coverage. In this simulation, each device received two CBR applications simultaneously.

For all simulations using the two CBR applications, the traditional device received both applications simultaneously through the 4G interface (generating competition between the applications) and when the traditional device detected the signal from the Wi-Fi network, it performed a handover to the new network and both applications started to be received simultaneously by the Wi-Fi interface (still generating competition between the applications).

When observing Figs 8–11 (first, second, third and fourth scenarios), the traditional device behaved less efficiently in all simulations, due to the use of both applications in the same wireless network interface, with an average bandwidth of 6 Mbps when connected to 4G. It is also possible to observe that in the traditional device there were drops in throughput (2 Mbps until the end of the simulations) in the period of 50 seconds of the simulations. This occurred because the traditional device had handed over the 4G network to the Wi-Fi network.

## First scenario

In this scenario for the proposed device, a CBR application was received through the 4G interface and another CBR application was received through the Wi-Fi interface, during the entire

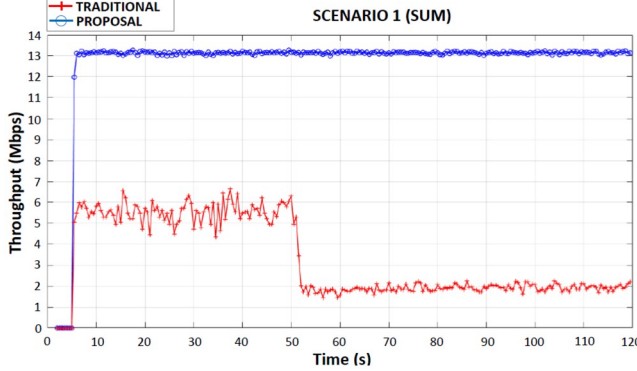

**Fig 9. First scenario throughput performance.**

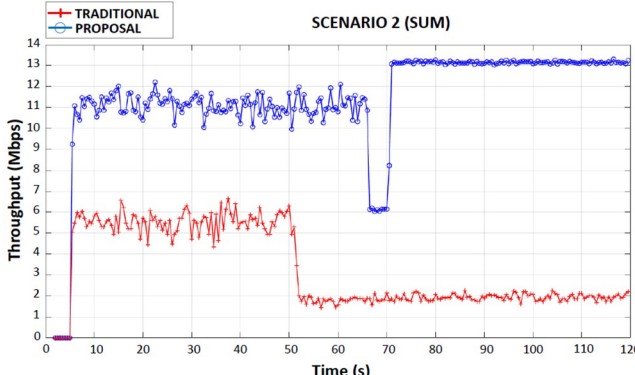

**Fig 10. Second scenario throughput performance.**

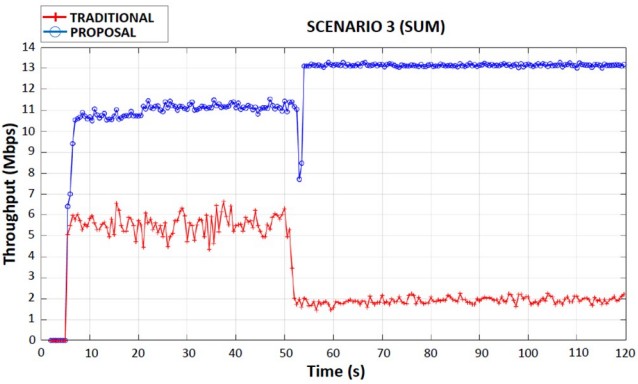

**Fig 11. Third scenario throughput performance.**

simulation the applications were connected simultaneously on both interfaces of wireless networks, thus avoiding competition between applications and consequently having a better throughput in relation to the traditional device (Fig 9).

During the simulation, the proposed device had a total average throughput of 13 Mbps (Fig 9 and Table 5), which is equivalent to the sum (since the applications were received simultaneously on different interfaces) of the throughput on the 4G interface, which was 7 Mbps with that of throughput on the Wi-Fi interface, which was 6 Mbps. The traditional device received both applications initially on the 4G interface and after the handover to the Wi-Fi interface the average flow rate dropped to 4 Mbps (total) during the entire transmission (Table 5).

## Second scenario

In this scenario, for the proposed device, a CBR application was received through the 4G interface and another CBR application was received through the Wi-Fi interface, during the entire

**Table 5. First scenario total average flow of traditional and proposed devices.**

| DEVICE | AVERAGE THROUGHPUT (Mbps) |
|---|---|
| Traditional | 4 |
| Proposed | 13 |

**Table 6. Second scenario total average flow of traditional and proposed devices.**

| DEVICE | AVERAGE THROUGHPUT (Mbps) |
|---|---|
| Traditional | 4 |
| Proposed (before Handover in Interface 4G) | 11 |
| Proposed (after Handover in Interface 4G) | 13 |

simulation the applications were connected simultaneously on both interfaces, thus avoiding competition between applications. However, in this simulation, the proposed device detected a new 4G base station and a new Wi-Fi access point, the possibility of handover for the two new networks detected was assessed by the Fuzzy System and it performed the handover only on the 4G interface to another base station 4G and kept the other interface connected to the same Wi-Fi access point.

In Fig 10, the proposed device had a total average throughput of 11 Mbps up to 70 seconds of the simulation time, which is equivalent to the sum of the throughput on the 5 Mbps on 4G interface with the throughput on the 6 Mbps on Wi-Fi interface. It is observed that, after the simulation reaches the time of 70 seconds, the proposed device only performed handover on the 4G interface (due to the rules of the Fuzzy System), which made it increase the average of the total throughput to 13 Mbps (sum of the flow rates of 7 Mbps of the 4G interface and 6 Mbps of the Wi-Fi interface), increasing the best use of the band, which proves the benefit of the proposal.

The traditional device received both applications initially on the 4G interface and after the handover to the Wi-Fi interface the average throughput rate dropped to 4 Mbps (total) during the entire transmission. Table 6 shows the average total throughput for the traditional device, for the proposed device before handover on the 4G interface and for the proposed device after handover on the 4G interface.

## Third scenario

In this scenario, for the proposed device, a CBR application was received through the 4G interface and another CBR application was received through the Wi-Fi interface. Again, the proposed device detected two new networks, a 4G network and a Wi-Fi network. The handover possibility for the two new networks detected was evaluated by the Fuzzy System. In this scenario, the proposed device remained connected to the current 4G network and performed handover to the new Wi-Fi network.

In Fig 11, the proposed device had a total average throughput of 11 Mbps up to 50 seconds of the simulation time, which is equivalent to the sum of the throughput on the 7 Mbps on 4G interface with the throughput of the 4 Mbps on Wi-Fi interface. It is observed that, after the simulation reaches the time of 50 seconds, the proposed device only performed handover to Wi-Fi interface to (due to the rules of the Fuzzy System), which made it increase the average total throughput to 13 Mbps (sum of the throughput rates of 7 Mbps from the 4G interface and 6 Mbps from the Wi-Fi interface), increasing the best use of the band, which proves the benefit of the proposal.

The traditional device received both applications initially on the 4G interface and after the handover to the Wi-Fi interface the average flow rate dropped to 4 Mbps (total) during the entire transmission. Table 7 shows the average total flow for the traditional device, for the proposed device before performing the handover on the Wi-Fi interface and for the proposed device after handover on the Wi-Fi interface.

Table 7. Third scenario total average flow of traditional and proposed devices.

| DEVICE | AVERAGE THROUGHPUT (Mbps) |
| --- | --- |
| Traditional | 4 |
| Proposed (before Handover in Interface Wi-Fi) | 11 |
| Proposed (after Handover in Interface Wi-Fi) | 13 |

## Fourth scenario

In this scenario, the device proposed precisely for receiving a CBR application on each interface continued to avoid competition between applications. In this scenario, the proposed device when detecting a new 4G network and a new Wi-Fi network, the Fuzzy System when analyzing the situation determined the realization of a double handover, that is, changing from the current 4G network to the new 4G network and changing from the current Wi-Fi network to the new Wi-Fi network.

Fig 12, the proposed device had a total average throughput of 10 Mbps up to 70 seconds of the simulation time, which is equivalent to the sum of the throughput on the 6 Mbps on 4G interface with the throughput on the 4 Mbps on Wi-Fi interface. It is observed that, after the simulation reaches a time of 70 seconds, the proposed device performed simultaneously handover the 4G to 4G and Wi-Fi to Wi-Fi interfaces to (due to the rules of the Fuzzy System), which caused the same to increase the average total throughput to 13 Mbps (sum of the throughput rates of 7 Mbps of the 4G interface and 6 Mbps of the Wi-Fi interface), which proves the benefit of the proposal.

The traditional device received both applications initially on the 4G interface and after the handover to the Wi-Fi interface the average throughput rate dropped to 4 Mbps (total) during the entire transmission. Table 8 shows the total average flow for the traditional device, for the

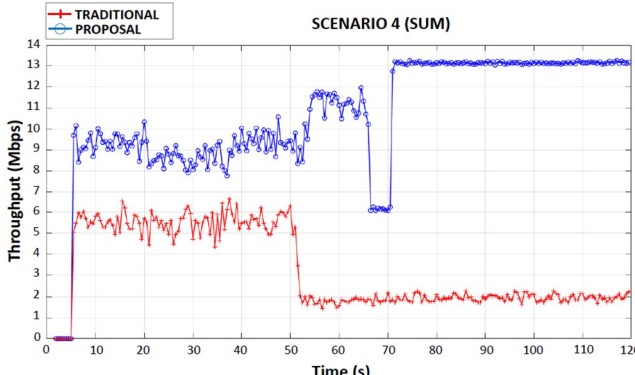

Fig 12. Fourth scenario throughput performance.

Table 8. Fourth scenario total average flow of traditional and proposed devices.

| DEVICE | AVERAGE THROUGHPUT (Mbps) |
| --- | --- |
| Traditional | 4 |
| Proposed (before Simultaneous Handover) | 10 |
| Proposed (after Simultaneous Handover) | 13 |

proposed device before performing the simultaneous handover on the 4G and Wi-Fi interfaces, and for the proposed device after simultaneous handover by the two wireless network interfaces.

## CBR and video application scenarios

The four scenarios described below were simulated again, but replacing a CBR application with a video application. Two applications were used for these simulations with the mobile devices, the first one was a CBR type application, with traffic throughput equivalent to 7 Mbps and the second, a video type application, (streaming), composed of 1999 frames.

In the traditional scenario, the mobile device received both application types—the CBR and the video—through the same interface (4G). Even after performing the handover, the mobile device continued to receive both applications on the new interface (Wi-Fi).

In the scenario with the implementation proposed, the mobile device received the video application through the 4G interface and the CBR application through the Wi-Fi interface, which avoided competition between technologies.

Still considering the proposed scenario, the following situations are likely to occur: 1) The mobile device remains connected to both 4G and Wi-Fi technologies without making a simultaneous handover of both technologies; 2) The mobile node remains connected to 4G and performs only the Wi-Fi handover; 3) The mobile node remains connected to Wi-Fi and performs only the 4G handover; 4) The mobile device performs simultaneous double handover from 4G to 4G and from Wi-Fi to Wi-Fi. In these scenarios, the evaluation was carried out using the QoE metrics: PSNR, SSIM and VQM.

## PSNR—Peak Signal to Noise Ratio

PSNR analyzes the error rate of the received video in relation to the original video, indicating the difference of the received frames in relation to the original frames. The PSNR value is expressed in dB (decibel) and for a video to be considered of good quality, it must have an average PSNR value of at least 30dB [11].

In the evaluation of PSNR in the simulations, Fig 13, the traditional device obtained an average PSNR value of 17dB, indicating that the video quality was of poor quality, as it did not reach the minimum value proposed by the QoE metric. For the proposed device (connected simultaneously on 4G and Wi-Fi networks) four results were evaluated. The first result evaluated was in a scenario in which there was no need to perform a handover on any interface, and the PSNR value was 42dB. The second result evaluated that the proposed device remained connected to the Wi-Fi network and performed a handover only on the 4G interface, with average

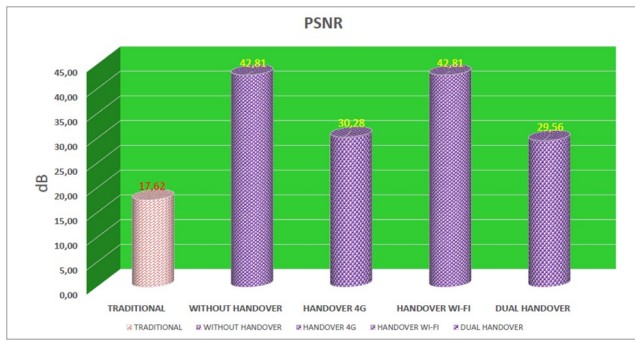

**Fig 13. Average PSNR of devices.**

PSNR of 30dB. The third result evaluated for the proposed device remained connected to the 4G network and performed a handover only on the Wi-Fi interface with an average PSNR value of 42dB. In the fourth evaluated result the proposed device performed a double handover, that is, simultaneous handover on 4G and Wi-Fi interfaces, in this case the average PSNE was 0.94.

## SSIM—Structural Similarity

SSIM evaluates the received video taking into account the color similarity, brightness and structure characteristics of the video. The SSIM value is expressed as a decimal value between 0 (zero) and 1, with values closer to 0 (zero) referring to a worse quality of the video [11].

In the evaluation of SSIM in the simulations, Fig 14, the traditional device obtained an average SSIM value of 0.67, indicating a lower quality of the video, as its value was more close to 0 (zero), according to the QoE metric. For the proposed device (connected simultaneously on 4G and Wi-Fi networks) four results were evaluated. The first result evaluated was in a scenario in which there was no need to perform a handover on any interface, and, the SSIM value was 0.97. The second result evaluated showed that the proposed device remained connected to the Wi-Fi network and performed a handover only on the 4G interface, obtaining SSIM of 0.95. The third result evaluated for the proposed device remained connected to the 4G network and performed a handover only on the Wi-Fi interface with an average SSIM value of 0.97. In the fourth evaluated result the proposed device performed a double handover, that is, simultaneous handover on 4G and Wi-Fi interfaces, in this case the average SSIM was 0.94.

## VQM—Video Quality Metric

VQM evaluates the distortion of colors, contrast, brightness, pixels, noise and, even, if the video is "blurred". The VQM value is expressed in a real number and the closer the value is to 0 (zero), the better the video quality will be [11].

In evaluating the VQM in the simulations, Fig 15, the traditional device obtained an average VQM value of 7.47, indicating poor video quality, as its value was farther from 0 (zero), according to the QoE metric. For the proposed device (connected simultaneously on 4G and Wi-Fi networks) four results were evaluated. The first result evaluated was in a scenario in which there was no need to perform a handover on any interface, in this case, the VQM value was 0.54. The second result evaluated showed that the proposed device remained connected to the Wi-Fi network and performed a handover only on the 4G interface, obtaining VQM of 1.08. The third result evaluated for the proposed device remained connected to the 4G network and performed a handover only on the Wi-Fi interface with an average VQM value of 0.54. In

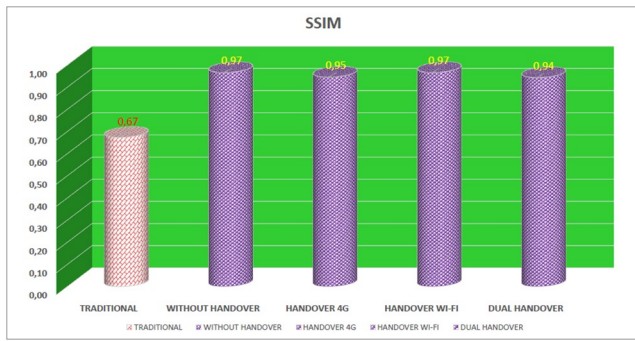

**Fig 14. Average SSIM of devices.**

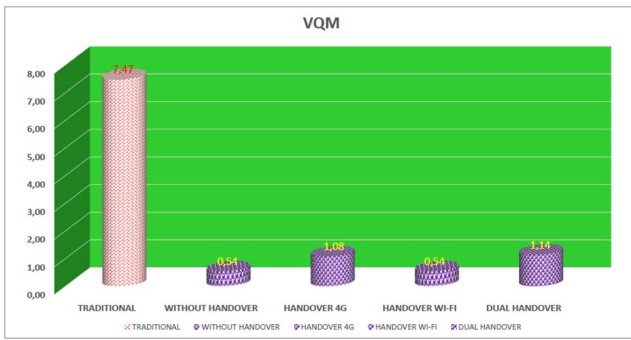

**Fig 15. Average VQM of devices.**

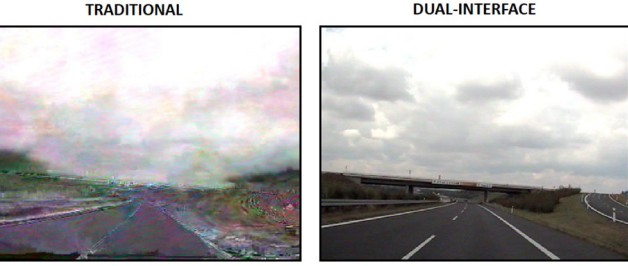

**Fig 16. Highway video frame 1139.**

the fourth evaluated result the proposed device performed a double handover, that is, simultaneous handover on 4G and Wi-Fi interfaces, in this case the average VQM was 1.14.

Fig 16 evidences the frames of the received videos compared both in the traditional scenario and in the dual-interface proposed. Because the interface of the traditional mobile device does not provide quality assurance for the two competing applications, the quality of the 1139th frame was distorted. An opposite situation occurred with the dual-interface mobile device: there was no distortion of frame 1139th and the quality of the transmitted video brought greater comfort to the user.

## Conclusion

With the exponential increase of mobile devices today, it is extremely important to observe the scalability of heterogeneous wireless networks, propose studies focused on the impacts caused by handover processes as well as on more effective solutions for the contribution to the maintenance of networks with different technologies. In view of this, the present study proposed the use of a mobile device that maintains multiple simultaneous connections to optimize the resources of available networks.

That being said, the present work demonstrated the use of a Fuzzy System to control the exchange of access points. This exchange should occur only when necessary to the user, and mainly intended to avoid ping pong handover, that is, performing handover to a new access point and then reconnecting back to the previous point.

The benefits stemming from our proposal also included an observable improvement in both QoS and QoE. For future research work, the proposal will include new technologies, as well as the implementation of new computational intelligence techniques.

## Acknowledgments

The authors would like to express they are thanks to the Federal University of the State of Pará and the Post-Graduate Electrical Engineering Program for its technical assistance and academic support. They are also grateful to the Coordinated Body for the Improvement of Higher Education Personnel (CAPES).

## Author Contributions

**Conceptualization:** Jorge Amaro de Sarges Cardoso, José Jailton Henrique Ferreira, Junior.

**Data curation:** Jorge Amaro de Sarges Cardoso, Fabio Pereira Ferreira da Silva.

**Formal analysis:** Jorge Amaro de Sarges Cardoso, Fabio Pereira Ferreira da Silva, José Jailton Henrique Ferreira, Junior.

**Investigation:** Jorge Amaro de Sarges Cardoso, Fabio Pereira Ferreira da Silva.

**Methodology:** Jorge Amaro de Sarges Cardoso, Fabio Pereira Ferreira da Silva, José Jailton Henrique Ferreira, Junior.

**Resources:** Jorge Amaro de Sarges Cardoso, Fabio Pereira Ferreira da Silva, José Jailton Henrique Ferreira, Junior.

**Supervision:** Tássio Costa de Carvalho, José Jailton Henrique Ferreira, Junior, Nandamudi Lankalapalli Vijaykumar, Carlos Renato Lisboa Francês.

**Writing – original draft:** Jorge Amaro de Sarges Cardoso, Fabio Pereira Ferreira da Silva, José Jailton Henrique Ferreira, Junior.

**Writing – review & editing:** Jorge Amaro de Sarges Cardoso, Tássio Costa de Carvalho, Nandamudi Lankalapalli Vijaykumar, Carlos Renato Lisboa Francês.

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
