## [Decision Letter · Decision Letter 0]

18 Nov 2020

PONE-D-20-29787

Heterogeneous Wireless Networks With Mobile Devices Of Multiple Interfaces For Simultaneous Connections Using Fuzzy System

PLOS ONE

Dear Dr. Cardoso,

Thank you for submitting your manuscript to PLOS ONE. After careful consideration, we feel that it has merit but does not fully meet PLOS ONE’s publication criteria as it currently stands. Therefore, we invite you to submit a revised version of the manuscript that addresses the points raised during the review process.

ACADEMIC EDITOR: The literature review requires revision to include more recent and relevant work in the literature. The propsoed design and the corresponding experiment setup require further elaborations and justifications. A minor revision is recommended. 

We look forward to receiving your revised manuscript.

Kind regards,

Chi-Tsun Cheng, Ph.D., M.Sc., B.Eng.

Academic Editor

PLOS ONE

Journal Requirements:

"No

The funders had no role in study design, data collection and analysis, decision to publish, or preparation of the manuscript.".

Additional Editor Comments (if provided):

The literature review requires revision to include more recent and relevant work in the literature. The propsoed design and the corresponding experiment setup require further elaborations and justifications. A minor revision is recommended.

Reviewers' comments:

Reviewer's Responses to Questions

**Comments to the Author**

1. Is the manuscript technically sound, and do the data support the conclusions?

Reviewer #1: Partly

Reviewer #2: Yes

2. Has the statistical analysis been performed appropriately and rigorously? 

Reviewer #1: Yes

Reviewer #2: I Don't Know

3. Have the authors made all data underlying the findings in their manuscript fully available?

Reviewer #1: Yes

Reviewer #2: Yes

4. Is the manuscript presented in an intelligible fashion and written in standard English?

Reviewer #1: Yes

Reviewer #2: Yes

5. Review Comments to the Author

Reviewer #1: To obtain more bandwidth and avoid competition between user applications on the same wireless technology interface，the manuscript proposes a dual interface device to realize simultaneous use of heterogeneous wireless networks, and a Fuzzy System to select the network for handover in homogeneous network. The simulation results are abundant and interesting, but it still has some problems.

1. The references on network selection are not up to date enough. The papers on network evaluation, selection and switching is scarce and insufficient.

2. The description of Fuzzy System is not detailed enough and lacks of references. The effect on reducing unnecessary handover has not been explained clearly from simulation results.

3. In this paper, the authors have mentioned the competition of different applications on the same wireless technology interface. Using two interfaces, the competition may be eliminated when running two applications. But what about the situations running three or more applications such as voice, video and web browsing? A scheme may need to be considered and further explained in this case.

4. Some sections have grammatical mistakes and should be carefully proofread and corrected.

Reviewer #2: The authors have produced a thorough investigation into the efficacy of dual interface heterogeneous access technology links for high data rate streaming, in particular for video links. The presented findings demonstrate the advantage of simultaneously connected heterogeneous access technologies in both link stability and achieving increased overall application level throughput.

It is noted that a comparison is made between a traditional mobile device capable of connecting only to a single access technology and a dual connectivity device with an intelligent handover (fuzzy logic) mechanism. However, it is unclear whether the results presented in figures 5,6,7,8 are representative of average performance or over a single run of the simulation. Further to this, although performance over the course of a significant period of time is presented, the statistical distribution of performance across multiple runs hasn't been shown in the case that the behaviour of the fuzzy system in particular is varied under probabilistic wireless channel conditions.

Similar concerns are expressed for figures 9,10 and 11 where 'average' performance is presented but it is not clear over how many simulations this average has been taken or whether it is the average performance over a single simulation. Likewise the distribution of these performances has not been indicated if a number of simulations have been performed.

Some minor issues are:

- Table 3 specifies a rate for each of the wireless access technologies in MB (Megabytes) where the interpretation of this may have been intended to be Mbps (Megabits per second).

- The spelling of 'base station' in Figure 1.

- There also appears to be a spelling error in Figure 12 with the spelling of traditional, possibly a mistranslation.

6. PLOS authors have the option to publish the peer review history of their article (what does this mean?). If published, this will include your full peer review and any attached files.

Reviewer #1: No

Reviewer #2: No

---

## [Author Response · Author response to Decision Letter 0]

4 Jan 2021

Dear Editor,

First of all, we would like to thank the reviewers for their availability and also to thank all the comments made and that we make every effort to contemplate all. We believe that after the changes made it has improved the quality of the paper.

Reviewer #1: To obtain more bandwidth and avoid competition between user applications on the same wireless technology interface，the manuscript proposes a dual interface device to realize simultaneous use of heterogeneous wireless networks, and a Fuzzy System to select the network for handover in homogeneous network. The simulation results are abundant and interesting, but it still has some problems.

1. The references on network selection are not up to date enough. The papers on network evaluation, selection and switching is scarce and insufficient.

R = New references were included in the related work, in particular, papers [9] e [10].

2. The description of Fuzzy System is not detailed enough and lacks of references. The effect on reducing unnecessary handover has not been explained clearly from simulation results.

R = A new Section was included for explaining Fuzzy. This should provide a better clarification on its functionalities. Also explained in details are possible answers to the systems.

3. In this paper, the authors have mentioned the competition of different applications on the same wireless technology interface. Using two interfaces, the competition may be eliminated when running two applications. But what about the situations running three or more applications such as voice, video and web browsing? A scheme may need to be considered and further explained in this case.

R = A new Section on Major Contributions was included. It explains ping pong handover reduction. Besides, a Figure (and due explanations) was included to show the solution with a middleware between two interfaces. Another Figure was also added to illustrate generic signaling on double handover. 

4. Some sections have grammatical mistakes and should be carefully proofread and corrected.

R = A native English speaker has corrected the mistakes and reviewed the entire article again to make the necessary corrections.

Reviewer #2: The authors have produced a thorough investigation into the efficacy of dual interface heterogeneous access technology links for high data rate streaming, in particular for video links. The presented findings demonstrate the advantage of simultaneously connected heterogeneous access technologies in both link stability and achieving increased overall application level throughput.

It is noted that a comparison is made between a traditional mobile device capable of connecting only to a single access technology and a dual connectivity device with an intelligent handover (fuzzy logic) mechanism. 

1. However, it is unclear whether the results presented in figures 5,6,7,8 are representative of average performance or over a single run of the simulation. 

R = The information was included in the text and in the Reference Parameter Table.

2. Further to this, although performance over the course of a significant period of time is presented, the statistical distribution of performance across multiple runs hasn't been shown in the case that the behaviour of the fuzzy system in particular is varied under probabilistic wireless channel conditions.

R = In the Major Contribution Section, a text is included to explain that one of the input parameters is RSSI of the interface that detected a new Access Point.

3. Similar concerns are expressed for figures 9,10 and 11 where 'average' performance is presented but it is not clear over how many simulations this average has been taken or whether it is the average performance over a single simulation. Likewise the distribution of these performances has not been indicated if a number of simulations have been performed.

R = This information was included in the text as well as in the Reference Parameters Table.

4. Some minor issues are:

- Table 3 specifies a rate for each of the wireless access technologies in MB (Megabytes) where the interpretation of this may have been intended to be Mbps (Megabits per second).

- The spelling of 'base station' in Figure 1.

- There also appears to be a spelling error in Figure 12 with the spelling of traditional, possibly a mistranslation.

R = All such minor issues were duly corrected.

---

## [Decision Letter · Decision Letter 1]

2 Feb 2021

Heterogeneous Wireless Networks With Mobile Devices Of Multiple Interfaces For Simultaneous Connections Using Fuzzy System

PONE-D-20-29787R1

Dear Dr. Cardoso,

We’re pleased to inform you that your manuscript has been judged scientifically suitable for publication and will be formally accepted for publication once it meets all outstanding technical requirements.

Kind regards,

Chi-Tsun Cheng, Ph.D., M.Sc., B.Eng.

Academic Editor

PLOS ONE

Additional Editor Comments (optional):

Most major issues have been resolved in the current revision. The paper is recommended to be accepted. However, the authors are highly recommended to check for typos in their final submission.

Reviewers' comments:

Reviewer's Responses to Questions

**Comments to the Author**

1. If the authors have adequately addressed your comments raised in a previous round of review and you feel that this manuscript is now acceptable for publication, you may indicate that here to bypass the “Comments to the Author” section, enter your conflict of interest statement in the “Confidential to Editor” section, and submit your "Accept" recommendation.

Reviewer #2: All comments have been addressed

2. Is the manuscript technically sound, and do the data support the conclusions?

Reviewer #2: Yes

3. Has the statistical analysis been performed appropriately and rigorously? 

Reviewer #2: Yes

4. Have the authors made all data underlying the findings in their manuscript fully available?

Reviewer #2: Yes

5. Is the manuscript presented in an intelligible fashion and written in standard English?

Reviewer #2: Yes

6. Review Comments to the Author

Reviewer #2: Following the comments made regarding the statistical nature of the simulations conducted and the way in which results have been presented, the authors have made changes that both clarify and increase the readability of the paper. The presentation of results now effectively demonstrates the efficacy of claims made in the discussion of said results.

The inclusion of the Major Contributions describes the application scenarios that are investigated in this paper and explains both the intended system architecture and operation.

A new section that describes the implemented fuzzy system also addresses concerns regarding system behaviour under probabilistic wireless channel conditions.

Most minor issues have been addressed. However, there remains a misspelt title in Figure 16 where "TRADICIONAL" should read "TRADITIONAL". Similarly in Figure 1. "BASE ESTATION" should be "BASE STATION" as written in Figure 2.

7. PLOS authors have the option to publish the peer review history of their article (what does this mean?). If published, this will include your full peer review and any attached files.

Reviewer #2: No

---

## [Editor Report · Acceptance letter]

9 Feb 2021

PONE-D-20-29787R1 

Heterogeneous Wireless Networks With Mobile Devices Of Multiple Interfaces For Simultaneous Connections Using Fuzzy System 

Dear Dr. Amaro de Sarges Cardoso:

I'm pleased to inform you that your manuscript has been deemed suitable for publication in PLOS ONE. Congratulations! Your manuscript is now with our production department. 

Kind regards, 

on behalf of

Dr. Chi-Tsun Cheng 

Academic Editor

PLOS ONE